# Prevalence and Risk Factors for Hyposalivation and Xerostomia in Childhood Cancer Survivors Following Different Treatment Modalities—A Dutch Childhood Cancer Survivor Study Late Effects 2 Clinical Study (DCCSS LATER 2)

**DOI:** 10.3390/cancers14143379

**Published:** 2022-07-11

**Authors:** Juliette Stolze, Jop C. Teepen, Judith E. Raber-Durlacher, Jacqueline J. Loonen, Judith L. Kok, Wim J. E. Tissing, Andrica C. H. de Vries, Sebastian J. C. M. M. Neggers, Eline van Dulmen-den Broeder, Marry M. van den Heuvel-Eibrink, Helena J. H. van der Pal, A. Birgitta Versluys, Margriet van der Heiden-van der Loo, Marloes Louwerens, Leontien C. M. Kremer, Henk S. Brand, Dorine Bresters

**Affiliations:** 1Princess Máxima Center for Pediatric Oncology, 3584 CS Utrecht, The Netherlands; j.c.teepen@prinsesmaximacentrum.nl (J.C.T.); j.l.kok-5@prinsesmaximacentrum.nl (J.L.K.); w.j.e.tissing@prinsesmaximacentrum.nl (W.J.E.T.); a.c.h.devries-15@prinsesmaximacentrum.nl (A.C.H.d.V.); m.m.vandenheuvel-eibrink@prinsesmaximacentrum.nl (M.M.v.d.H.-E.); h.j.h.vanderpal@prinsesmaximacentrum.nl (H.J.H.v.d.P.); a.b.versluijs@prinsesmaximacentrum.nl (A.B.V.); m.vanderheiden@prinsesmaximacentrum.nl (M.v.d.H.-v.d.L.); l.c.m.kremer@prinsesmaximacentrum.nl (L.C.M.K.); d.bresters@prinsesmaximacentrum.nl (D.B.); 2Department of Oral Biochemistry, Academic Center for Dentistry Amsterdam (ACTA), 1081 LA Amsterdam, The Netherlands; h.brand@acta.nl; 3Department of Oral Medicine, Academic Center for Dentistry Amsterdam (ACTA), 1081 LA Amsterdam, The Netherlands; judith@raber.nl; 4Department of Oral and Maxillofacial Surgery, Amsterdam University Medical Center (UMC), Location AMC, 1105 AZ Amsterdam, The Netherlands; 5Radboud University Medical Center, 6525 GA Nijmegen, The Netherlands; jacqueline.loonen@radboudumc.nl; 6Department of Pediatric Oncology, Beatrix Children’s Clinic, University Medical Center Groningen, 9713 GZ Groningen, The Netherlands; 7Department of Pediatric Oncology, Sophia Children’s Hospital, Erasmus Medical Center, 3015 GD Rotterdam, The Netherlands; 8Department of Internal Medicine, Section Endocrinology, Erasmus Medical Center, 3015 GD Rotterdam, The Netherlands; s.neggers@erasmusmc.nl; 9Emma Children’s Hospital, Amsterdam UMC, Location VUmc, 1105 AZ Amsterdam, The Netherlands; eline.vandulmen-denbroeder@vumc.nl; 10Department of Internal Medicine/Endocrinology, Leiden University Medical Center, 2333 ZA Leiden, The Netherlands; m.louwerens@lumc.nl; 11Wilhelmina Children’s Hospital, University Medical Center Utrecht, 3584 EA Utrecht, The Netherlands; 12Emma Children’s Hospital, Amsterdam UMC, Location AMC, 1105 AZ Amsterdam, The Netherlands

**Keywords:** childhood cancer, late effects, decreased salivary flow rate, hyposalivation, salivary gland dysfunction, xerostomia, cancer survivors, oral health

## Abstract

**Simple Summary:**

Salivary gland dysfunction is an underestimated late effect in childhood cancer survivors (CCS). The objective of this cross-sectional study, part of the multidisciplinary multicenter Dutch Childhood Cancer Survivor Study Late Effects 2 (DCCSS LATER 2), was to assess the prevalence of and risk factors for hyposalivation and xerostomia in CCS with a long-term follow-up exceeding 15 years. From February 2016 until March 2020, 292 CCS were included. The prevalence of hyposalivation was 32% and the prevalence of xerostomia was 9.4%. Hyposalivation and xerostomia did not correlate significantly. Risk factors for hyposalivation were female gender and a higher dose of radiotherapy (>12 Gy) to the salivary glands. Screening for hyposalivation during long-term follow-up in CCS is recommended in order to provide optimal oral supportive care aimed to improve oral health.

**Abstract:**

Background: Limited data are available on the risk factors of salivary gland dysfunction in long-term childhood cancer survivors (CCS). The objective of this cross-sectional study, part of the multidisciplinary multicenter Dutch CCS Study Late Effects 2 (DCCSS LATER 2), was to assess the prevalence of and risk factors for hyposalivation and xerostomia in CCS. Methods: From February 2016 until March 2020, 292 CCS were included. Data with regard to gender, age at study, diagnosis, age at diagnosis, and treatment characteristics were collected, as well as the unstimulated (UWS) and stimulated whole salivary flow rate (SWS). Xerostomia was assessed with the Xerostomia Inventory (XI) questionnaire. Multivariable Poisson regression analyses were used to evaluate the association between potential risk factors and the occurrence of hyposalivation. Results: The minimum time between diagnosis and study enrollment was 15 years. The prevalence of hyposalivation was 32% and the prevalence of xerostomia was 9.4%. Hyposalivation and xerostomia were not significantly correlated. Risk factors for hyposalivation were female gender and a higher dose of radiotherapy (>12 Gy) to the salivary gland region. Conclusion: Considering the importance of saliva for oral health, screening for hyposalivation in CCS is suggested in order to provide optimal oral supportive care aimed to improve oral health.

## 1. Introduction

Over 75% of childhood cancer survivors (CCS) experience one or more late effects arising from childhood cancer treatment [1]. Amongst these late effects, survivors may develop salivary gland dysfunction, such as hyposalivation (decreased salivary secretion) and/or xerostomia (the subjective feeling of a dry mouth) [2,3,4]. Salivary gland dysfunction is a significant and probably underestimated late effect and may negatively affect general health [5,6], as saliva maintains oral health by protecting the oral mucosa and teeth. Complications arising from hyposalivation include oral discomfort and pain, difficulty speaking, mastication and swallowing, decreased taste perception, nutritional compromise, sleep disorders, dental erosion, xerostomia, increased prevalence of dental caries, and susceptibility to oral mucosal infections [7,8]. These complications can significantly affect oral health-related quality of life [9,10].

Risk factors for salivary gland dysfunction among cancer patients and survivors include head and neck radiotherapy [4,11], conditioning for hematopoietic stem cell transplantation (HSCT) with total body irradiation [8], graft-versus-host-disease [12], and female gender [13,14]. An increased prevalence of salivary gland dysfunction among CCS has been reported [4]. However, limited data on risk factors, such as the impact of radiotherapy dosage or different types of chemotherapy, are available, and no studies in large cohorts of CCS with a long follow-up and with complete data on diagnosis and treatment have been performed. Therefore, in this study, we aimed to assess the prevalence of hyposalivation and xerostomia in CCS with a long-term follow-up exceeding 15 years. In addition, we attempted to define risk groups among CCS for hyposalivation and xerostomia. With this information, long-term follow-up guidelines on oral care in CCS can be improved in order to provide optimal oral supportive care and to prevent oral diseases.

## 2. Methods

### 2.1. Design

This cross-sectional study is part of the SALI (abbreviation: hypoSALIvation) subproject, part of the multidisciplinary Dutch Childhood Cancer Survivor Study (DCCSS) LATER 2 study. The SALI study was approved by the Medical Ethical Committee of the Amsterdam University Medical Center, the Netherlands (protocol number MEC2013_127). Informed consent was obtained from all subjects.

### 2.2. Patients

In the DCCSS LATER 2 study, CCS were included from February 2016 until March 2020. Survivors were eligible for inclusion if diagnosed with childhood cancer between 1963 and 2001 in one of the seven pediatric oncology centers in the Netherlands before the age of 18 years and have survived at least 5 years since diagnosis of the malignancy. This nationwide cohort of more than 6000 survivors was described elsewhere (Teepen et al., manuscript submitted). In the SALI subproject, participants were included from three of the seven outpatient clinics of DCCSS LATER 2: Amsterdam University Medical Center (UMC) location VUmc, Leiden University Medical Center (LUMC), and Princess Máxima Center for Pediatric Oncology (PMC).

### 2.3. Study Groups

Since our hypothesis was that radiotherapy and chemotherapy would affect long-term salivary gland functioning to a different extent, during inclusion for the SALI subproject, we aimed to create three study groups: a group of CCS who received head and neck radiotherapy (H&N RT) or total body irradiation (TBI) without chronic graft versus host disease (cGVHD), a group of CCS with (a history of) cGVHD after HSCT, and a group of CCS treated with chemotherapy and no H&N RT or TBI. Inclusion in the group with cGVHD did not reach the minimum number of 40 participants needed for analysis as a group, although the three participating outpatient clinics included the majority of CCS who had undergone HSCT and were thus at risk for cGVHD. Therefore, this group was omitted from the study, resulting in two groups for analysis: Group 1: CCS who did not receive H&N RT (but all received chemotherapy), and Group 2: CCS who received H&N RT and/or TBI. The few included participants in the initial group with cGVHD were included in either Group 1 or Group 2 based on the type of therapy. For statistical analysis, Group 2 was further subdivided into two subgroups: subgroup 2A, those who received RT in a field involving salivary glands, and subgroup 2B, without direct in-field exposure of salivary glands to irradiation.

### 2.4. Data Collection

Data with regard to gender, age at study, primary cancer type, age at diagnosis, and treatment characteristics were collected by study data managers using a standardized protocol. Treatment exposure data covered treatment for the initial childhood tumor, all recurrences, and any new malignancy. Patients who received H&N RT (Group 2) were further subdivided into whether they did (subgroup 2A) or did not (subgroup 2B) receive radiotherapy to the salivary glands based on coded radiation fields in the LATER data registry. The radiation fields concerned full brain RT, partial brain RT, the face, and the neck. Whether salivary glands were within these fields of radiation was determined by the use of anatomy and literature on the location of the salivary glands. In CCS who received RT to the face (including the sublingual and submandibular glands and possibly the parotid glands) and/or the central part of the head (including the parotid glands), radiotherapy was classified as “radiotherapy to salivary gland”. In CCS who received partial brain RT, it was unclear whether salivary glands were in the field of irradiation, thus radiotherapy was classified as “radiotherapy to salivary glands unclear”. Radiotherapy to the neck was classified as “radiotherapy not to salivary glands”. If information about the radiation fields was unclear, these CCS were not included in one of the two subgroups (2A or 2B) and, thus, were excluded from the analysis. The total dose to the salivary gland region was calculated based on the prescribed radiation dose. In case a patient received multiple radiotherapy treatments, the prescribed dose was summed for overlapping fields. For non-overlapping fields, the highest maximum prescribed dose of one of the non-overlapping fields was assigned. Until the 1990s, two-dimensional radiotherapy (2D-RT) was the standard of care [15,16,17]. This included, e.g., conventional techniques based on two opposing beams; the definitions of the target volume and dose calculation were not completely based on computed tomography scans. These scenarios typically concerned whole cranium and craniospinal axis fields for leukemia [18] and medulloblastoma [19]. Before the 1990s, total body irradiation (TBI) was given in 1 fraction of 7.5 or 8 Gy or 2 fractions of 5 or 6 Gy on consecutive days with conventional techniques (2D-RT) [20]. Since the 1990s, CT-based three-dimensional radiation treatment (3D-RT) and multi-beam radiation delivery techniques have been implemented in pediatric oncology, sometimes referred to as ‘conformal radiotherapy’ [21]. Compared to 2D-RT, the dose from 3D-RT delivered to healthy tissues surrounding the tumor in the high-dose region is generally lower at the cost of a larger area of healthy tissues receiving low-dose irradiation. Chemotherapy data included drug names for all chemotherapeutic agents. The following groups of chemotherapy were created based on the specific agents: alkylating agents, vinca alkaloids, epipodophyllotoxins, anthracyclines, platinum compounds, and antimetabolites.

Study personnel of the participating centers was trained in salivary flow assessment procedures. During the study visit in the clinic, unstimulated (UWS) and stimulated whole salivary flow rates (SWS) were measured according to internationally standardized procedures. For the collection of UWS, participants were asked to first swallow and thereafter spit all accumulated saliva into a plastic container every 30 s for 5 min. For the collection of SWS, participants were asked to repeat this while chewing on a 5 × 5 cm piece of tasteless Parafilm (Bemis Inc, Neenah, GA, USA). Salivary flow rates were determined gravimetrically and expressed as mL/min [22]. The UWS and SWS were categorized into ‘hyposalivation’ (<0.2 mL/min and <0.7 mL/min, respectively) and ‘severe hyposalivation’ (<0.1 mL/min and <0.5 mL/min, respectively) [23,24]. To determine the severity of xerostomia, participants were asked to fill out the Dutch translation of the Xerostomia Inventory (XI) [25]. This internationally validated questionnaire consists of 11 items with a 5-point Likert scale (1 = never to 5 = very often). The sum of the responses to the 11 items results in an XI score that ranges from 11 (no dry mouth) to 55 (extremely dry mouth). Participants with scores above 25 were classified as participants who experienced xerostomia. 

All participants in the DCCSS LATER study were also asked to fill out an extensive questionnaire including questions on their use of any medication if used at least once a week. The total number of medicines was calculated for each participant. Medication was excluded if it was not registered in the Dutch national database of prescribed medications by general practitioners or medical specialists (Farmacotherapeutisch Kompas, NL, USA), such as multivitamins or homeopathic preparations. We classified participants as having polypharmacy when 4 or more medicines were used, in accordance with the study by Assy et al. [26].

### 2.5. Statistical Analyses

Patient demographics and disease- and treatment characteristics were summarized using descriptive statistics and compared according to treatment modality using the Mann–Whitney U test for continuous variables and Fisher’s exact test for categorical variables. Frequencies of xerostomia and mean and median values of XI scores, frequencies of hyposalivation for both UWS and SWS, and mean and median values of UWS and SWS were reported and compared by treatment modality and gender using Mann–Whitney U and Fisher’s exact tests. The associations between UWS, SWS, and xerostomia were explored with the chi-squared test. Univariable Poisson regression analyses were performed between potential risk factors and occurrences of hyposalivation measured by UWS and SWS. Multivariable Poisson regression analyses with the log–link function and robust standard errors to calculate relative risks were used to evaluate the association between potential risk factors and the occurrence of hyposalivation for UWS and SWS, separately [27]. In both models, we included known potential risk factors based on the literature (radiotherapy to the salivary gland region, gender [4,13,14]) and both models were adjusted for age at diagnosis and time since diagnosis to account for differences in age and time since treatment exposures. Other potential risk factors (radiotherapy dose to salivary gland region, type of chemotherapy (assessed per group: anthracyclines, epipodophyllotoxins, anti-metabolites, vinca alkaloids, alkylating agents, and platinum compounds), polypharmacy) were only included in the multivariable model when the variable had a *p*-value < 0.1 in the univariable Poisson regression analysis. Testing for trends of the radiotherapy dose to the salivary gland region was based on the likelihood-ratio-based *p*-value for a model with the relevant continuous variable on the basis of exposed patients only. IBM SPSS version 26.0 (IBM Corp. Armonk, NY, USA) was used to perform data analyses.

## 3. Results

### 3.1. Patient Characteristics

Of the 617 invited CCS, a total of 292 participated in the study and could be included in the analysis. Figure 1 provides a flowchart of the inclusion process. Table 1 shows the characteristics of the included CCS. There was an almost equal distribution between men (53.1%) and women (46.9%). A majority of the survivors were diagnosed with hematological malignancy (74.0%) (Table 1, Appendix A) [28]. The median age at cancer diagnosis was 5.20 years (range: 0.01–17.00 years) for the total study population. The minimum follow-up time between cancer diagnosis and enrollment in the study was 15.94 years, with a median time (since diagnosis) of 25.26 years. A total of 198 CCS (67.8%) were included in Group 1: no H&N RT; 94 CCS (32.3%) were included in Group 2: H&N RT.

Compared to Group 1, in Group 2 (H&N RT), a greater proportion of CCS were men (62.8% vs. 48.5%), and the median age at cancer diagnosis was significantly higher (median 7.47 years vs. 3.97 years) and follow-up time was longer (median 32.61 vs. 23.96 years). Compared to Group 1, in the H&N RT group, a higher number of CCS received allogeneic HSCT (23.4% vs. 8.1%). In Group 1, all CCS received chemotherapy and 10.1% received radiotherapy in fields other than H&N. In the H&N RT group, 88.3% also received chemotherapy, whereas 86.2% received radiotherapy to the salivary glands. Table 2 shows the distribution of radiation fields to different H&N areas and the mean and median dose applied.

### 3.2. Xerostomia and Salivary Flow Rates

The data on xerostomia and salivary flow rates are presented in Table 3, stratified for different treatment modalities and gender. Of the 292 participating CCS, 233 (79.8%) filled out the Xerostomia Inventory. Among these 233 CCS, 9.4% experienced xerostomia. Percentages of CCS with xerostomia did not differ significantly between CCS who did not receive H&N RT (Group 1) and CCS who received RT to salivary glands (Group 2A), nor between men and women. UWS values were available for 269 of the 292 CCS (92.1%) and ranged from 0.00 to 1.42 mL/min. SWS values were measured for 271 of the 292 CCS (92.8%) and ranged from 0.01 to 4.03 mL/min. Among these CCS, hyposalivation was identified in 32.0% (based on UWS) and 31.7% (based on SWS), which was severe in 8.9% (UWS) and 14.0% (SWS). The prevalence of hyposalivation in CCS as assessed by both UWS (*p* = 0.002) and SWS (*p* = 0.004) was significantly higher among those with a history of irradiation to salivary glands (Subgroup 2A), compared to CCS who did not receive H&N RT. UWS and SWS values were significantly lower (*p* = 0.003) among CCS who received RT to salivary glands than among CCS who did not receive H&N RT. Among women, the prevalence of hyposalivation in CCS based on both UWS (*p* = 0.019) and SWS (*p* = 0.018) and the prevalence of severe hyposalivation based on SWS (*p* = 0.013) was significantly higher than among men. Compared with men, women had significantly lower median values for both UWS (*p =* 0.007) and SWS (*p* = 0.001).

Table 4 shows the association between unstimulated and stimulated hyposalivation and between hyposalivation and xerostomia. The presence of hyposalivation based on UWS and SWS was significantly associated (*p* < 0.001). In contrast, xerostomia was not significantly associated with objectively measured hyposalivation.

### 3.3. Risk Factor Analysis

Appendix A shows univariable Poisson regression analyses between potential risk factors and the prevalence of hyposalivation. The results of the Poisson multivariable regression analysis on the roles of patient- and treatment-related characteristics in the prevalence of hyposalivation are presented in Table 5 (UWS) and Table 6 (SWS).

For hyposalivation based on UWS, the number of medications was associated with hyposalivation in univariable analysis (Appendix A) and, therefore, included in the multivariable model in addition to potential risk factors based on the literature (radiotherapy to the salivary gland region and gender [4,13,14]) (Table 5). The relative risk of hyposalivation (based on UWS) was associated with female gender (RR women vs. men, 1.52; 95% CI, 1.06 to 2.19) and a longer time since diagnosis (RR per 10 years, 1.42; 95% CI, 1.15 to 1.75). Furthermore, CCS who received RT to salivary glands with a prescribed dose higher than 34 Gy had a significantly increased risk of hyposalivation (UWS) compared to survivors who did not receive RT to salivary glands (RR > 34 Gy versus 0 Gy, 2.10; 95% CI, 1.21 to 3.63) (*p trend for dose* = 0.384). For hyposalivation based on SWS, treatments with vinca alkaloids, alkylating agents, and anthracyclines were associated with hyposalivation in univariable analyses (Appendix A) and, therefore, included in the multivariable model in addition to the potential risk factors based on the literature (Table 6). In the multivariable analyses, women had a significantly higher risk of hyposalivation (SWS) than men (RR women vs. men, 1.57; 95% CI, 1.10 to 2.23). Radiotherapy to salivary glands region > 12 Gy was associated with an increased risk of hyposalivation (SWS), with RRs of 1.74 (95% CI, 0.95 to 3.20), 2.15 (95% CI, 1.26 to 3.67) and 2.25 (95% CI, 1.35 to 3.76) for >0 and ≤12 Gy, >12 and ≤34 Gy, and >34 Gy, respectively, compared to patients without radiotherapy to the salivary gland region (*p trend* = 0.832). In the multivariable analysis, vinca alkaloids (RR, 0.67; 95% CI, 0.44–1.02), anthracyclines (RR, 0.81; 95% CI,0.56–1.18), and alkylating agents (RR, 0.81; 95% CI, 0.55–1.19) were not significantly associated with hyposalivation based on SWS.

## 4. Discussion

To our knowledge, this is the first study to assess the prevalence of and risk factors for hyposalivation and xerostomia in a relatively large cohort of childhood cancer survivors with a long follow-up time (median 25.3 years after diagnosis). Among the CCS in the present study, the prevalence of objectively measured hyposalivation was 32% and the prevalence of xerostomia, the sensation of dry mouth, was 9.4%. The presence of hyposalivation and xerostomia were not significantly associated. The main risk factors for hyposalivation in CCS were female gender and a higher (>12 Gy) prescribed dose of radiotherapy to the salivary gland region.

### 4.1. Hyposalivation

#### 4.1.1. Prevalence

Overall, it is difficult to compare salivary flow rates with the general population, as salivary flow rates vary considerably due to different factors, such as gender, age, diseases, and polypharmacy. Several studies compared salivary secretion rates between children who received cancer therapy and the controls [29,30,31]. Those studies found that UWS rates did not differ significantly between survivors and controls [29,30], but SWS rates were significantly lower in CCS [30,31]. However, as these measurements took place at the age of 12 years and secretion rates reach normal adult values at the age of 15 [32], we could not meaningfully compare our results to those studies.

Several other studies reported on salivary flow rates in adult general populations. For UWS, flow rates in our study were slightly higher for both genders than in a study among a random Swedish population of different age groups ranging between 20 and 69 years (men: mean 0.39 mL/min vs. 0.33 mL/min, women 0.30 mL/min vs. 0.26 mL/min) [33], but slightly lower compared to another study among healthy adults between 35 and 64 years (UWS rates ranging between 0.42 and 0.58 mL/min in men and between 0.20 and 0.46 mL/min in women) [34]. The mean SWS rates in the present study were clearly lower than the rates found in the aforementioned Swedish population study (1.27 mL/min versus 2.50 mL/min in men and 0.98 mL/min versus 2.03 mL/min in women), and the prevalence of a low stimulated salivary flow rate (<0.7 mL/min) was clearly higher (32.0% versus 2.8%) [33]. A possible explanation for the SWS being more affected than the UWS could be that among CCS in the present study, parotid glands were more often in the radiation field (full brain RT and TBI) than submandibular and sublingual glands (RT to the face and TBI) and, therefore, likely more often affected. The parotid glands primarily provide stimulated salivary secretion. 

Studies reporting the prevalence of salivary gland dysfunction in CCS are scarce and, in contrast to the present study, only focused on HSCT recipients [3,13,14,35,36,37]. In these studies, prevalence of hyposalivation was measured by UWS (<0.1 mL/min) after a follow-up of 7 years since HSCT varied between 6% and 54% [3,35,37]. Among CCS in the present study with a median follow-up of 25.3 years, hyposalivation (<0.1 mL/min) was present in 8.9%. Studies that based hyposalivation on a SWS < 0.5 mL found prevalence numbers ranging from 24% to 70% after a 1 year follow-up [13,36], 8% to 43% after 4 years [38], 26% to 54% after 7 years [3,35,37], and 42% to 47% after 8 years [14]. Based on a cut-off value of SWS < 0.7 mL/min, a prevalence of 53% was reported [35]. The difference in prevalence as compared to our study (31.7%, <0.7 mL/min; 14.0%, <0.5 mL/min) may be explained by the lower percentage of CCS who received HSCT (16.8%) and TBI (9.9%) in our study.

#### 4.1.2. Risk Factors

Head and neck radiotherapy is a known risk factor for hyposalivation in adults [8].

To our knowledge, no previous data are available on decreased salivary flow rates in CCS treated with different prescribed doses of radiotherapy to salivary glands, except for several studies reporting UWS and SWS rates among recipients of pediatric HSCT with a mean follow-up time varying between 1 and 8 years, after a conditioning regimen with TBI or chemotherapy. Two studies reported significantly lower UWS and SWS among recipients of HSCT and a conditioning regimen with single-fraction (sTBI) as compared to fractionated TBI (fTBI) [36] and among recipients of HSCT and a conditioning regimen with sTBI/cyclophosphamide(Cy) as compared to chemotherapy only (busulfan/Cy) [35]. In three studies, a significantly lower SWS was reported in recipients conditioned with TBI vs. chemotherapy only [13,38,39]. Two other studies did not find significant differences in UWS rates and SWS rates between conditioning regimens with or without TBI [3,14]. In HSCT patients with a TBI-based conditioning regimen as compared to age- and gender-matched controls, significantly lower SWS rates were reported [38,39]. In another study, significantly lower UWS and SWS rates were found after HSCT and a conditioning regimen with TBI as compared to pre-HSCT measurements [37].

By contrast, in the present study, TBI did not significantly increase the risk for hyposalivation (the group with radiotherapy dosage to salivary glands of >0 and ≤12 Gy), whereas a higher dose of RT to salivary glands did. This is in accordance with a systematic review [8], which discusses that damage to salivary gland tissue depends on the cumulative dose of irradiation. A higher radiation dose per fraction with a lower cumulative dose, such as TBI, results in less damage to salivary gland tissue as compared to lower radiation doses per fraction with higher cumulative doses, such as H&N radiotherapy [8].

In the present study, radiotherapy seems to have a more pronounced negative effect on hyposalivation based on SWS than UWS. This can be explained, because the parotid glands, the main contributors to stimulated salivary flow [40], were most often in the field of radiotherapy (full brain RT and TBI, *n* = 73), as compared to the sublingual and submandibular glands (face RT and TBI, *n* = 37). Modern techniques, such as intensity-modulated radiotherapy (IMRT), make it possible to spare the parotid glands from irradiation [8] or spare certain regions of the parotid glands containing stem cells [41], thereby reducing post-radiotherapy salivary gland dysfunction. Still, it is evident that radiotherapy affects both unstimulated and stimulated salivary flow rates in long-term survivors of childhood cancer and, therefore, contributing glands of resting saliva production must also be spared from radiotherapy, if possible, as recommended by a recent systematic review [4].

Female gender was shown to be another risk factor for hyposalivation. This is in line with several studies, which reported significantly lower salivary secretion rates and higher occurrences of hyposalivation among women compared to men after pediatric HSCT [13,14,35,38] and after HSCT in adults [42]. Moreover, in the healthy general population, women have significantly [33,34,43,44] lower mean UWS flow rates than men, which is suggested to be due to smaller gland sizes in women compared to men [43], and/or hormonal changes [34].

Due to the cross-sectional design of the present study, salivary flow rates were measured at one time point and the precise effect of age on changes in salivary flow rates within the survivor cohort could therefore not be examined. One study found that a higher age correlated with decreased salivary flow rates [34], possibly due to age-related diseases and increased use of medication at older ages [45]. In the present study, for each 10-year increase in time since childhood cancer diagnosis (a proxy for age), in a multivariate analysis, there was a 1.45-fold higher risk of hyposalivation based on UWS; however, time since childhood cancer diagnosis was not found to be a risk factor for hyposalivation based on SWS. Similarly, in Dodds et al., unstimulated salivary flow rates from the submandibular and sublingual glands showed the most significant age-related decline in flow rates in a community-based cohort with a mean age of 59.9 years (range 35–81) [34]. Although some studies found associations between usage of multiple medications and decreased salivary flow rates [26,33], in the present study, the number of medications, irrespective of their xerogenic potential, was not found to be a risk factor for hyposalivation. Xerogenic medication is associated with oral dryness [46], and thereby we may have underestimated the actual effect of known xerogenic medicines in CCS.

### 4.2. Xerostomia

Prevalence of xerostomia differed between studies, which is likely due to differences in the manner by which xerostomia was assessed. The prevalence of xerostomia in our study (9.4% in the total group, 11.7% in those who received H&N RT) was lower than reported in a recent systematic review, which suggested that the incidence of acute (subjective) xerostomia (RTOG EORTC) in CCS who received H&N RT with a mean dose of 35 to 40 Gy is approximately 32%, and chronic grade 2 xerostomia occurs in 13% to 32% of these patients [4]. In our study, the Xerostomia Inventory was used, an internationally validated questionnaire with 11 items to quantify the severity of xerostomia [25], while in other studies, xerostomia was graded according to the RTOG/EORTC using only four grades or according to the CTCAE using five grades [47,48]. In another study among 8522 CCSs with a median time of 22 years between diagnosis and study enrollment, compared with a sibling cohort, in the multivariate analysis, survivors were significantly more likely to report xerostomia; 2.8% vs. 0.3% (OR of 9.7, 95% CI 4.8–19.7, *p* < 0.01) [49]. In that study, xerostomia was assessed with a single question regarding difficulty producing saliva that required treatment, such as artificial saliva [49]. Among 53 recipients of allogeneic HSCT, with a mean follow-up time of 6.9 (sd 4.0) years and a mean age of 13.4 (sd 4.0) years at examination, 79% of the participants expressed complaints of xerostomia and 49% expressed two or more dry mouth-related complaints [3]. In this study [3], xerostomia was assessed as having positive answers to nine questions indicating dry mouth, summed to determine the xerostomia score. The study found that the xerostomia score was inversely correlated to UWS and SWS (*p* < 0.01) [3].

In the present study, however, symptoms of xerostomia did not correlate with the occurrence of (objectively measured) hyposalivation. Only a small group of the participants who had an objectively decreased salivary secretion rate experienced xerostomia (10% to 20%). This is an important finding, indicating that not all CCS recognize their actual decreased salivary secretion rates. This may be related to the composition of saliva. Xerostomia merely affects oral health-related quality of life, whereas hyposalivation directly affects oral health by increasing the risk of caries, gingival inflammation, oral discomfort and pain, dental erosion, and oral mucosal infections. According to our results, it appears that CCS are unaware of the presence of hyposalivation. As hyposalivation is prevalent among CCS and negatively affects oral health, it deserves more attention. This is especially important in CCS, because hyposalivation and xerostomia may develop already at a young age and, therefore, they may not appreciate their oral mouth as abnormal and may not complain of xerostomia. Thus, we recommend that any evaluation of dry mouth should not be limited to the assessment of subjective complaints, but should also include measurements of salivary secretion rates.

### 4.3. Strengths and Limitations

A strength of the present study is that both subjective symptoms of xerostomia, as well as objectively decreased salivary flow rates, were measured using validated questionnaires and standardized saliva collection procedures. Moreover, detailed information about the different treatment modalities and doses was collected. In addition, we achieved a long follow-up time of more than 15 years, by which we were able to capture the long-term effects. 

A limitation of the study is that we did not have data on xerostomia and salivary flow rates of a control group of non-cancer patients (e.g., siblings). However, because we used validated questionnaires and standardized measurements, we were able to compare our results with other studies. 

The SALI subproject selected CCS who had received H&N RT or TBI, and CCS who had not. As this cohort is not representative of the Dutch CCS cohort as a whole, the overall prevalence of outcomes should be interpreted with caution. On the other hand, this selection procedure allowed us to properly investigate the prevalence of a low salivary flow rate and xerostomia between CCS who had received H&N RT or TBI and those who had not. With a response rate of 50% of the CCS eligible for this study, a possible inclusion bias could exist, as participants with more serious oral problems/xerostomia may have been more likely to participate in the study. 

## 5. Conclusions

The prevalence of hyposalivation in childhood cancer survivors is relatively high, with one-third of the included survivors having objectively measured hyposalivation more than 15 years after diagnosis. The present study shows that female gender and a higher prescribed dose of radiotherapy to the salivary glands are the main risk factors for hyposalivation. Hyposalivation was not significantly associated with the subjective feeling of dry mouth, indicating that not all CCS recognize their actual decreased salivary secretion rate. Therefore, to reveal this prevalent late effect, screening of salivary gland dysfunction during long-term follow-ups in CCS is recommended in order to provide optimal oral supportive care aimed to improve oral health. We suggest that late effect screening guidelines could be adjusted accordingly.

## Figures and Tables

**Figure 1 cancers-14-03379-f001:**
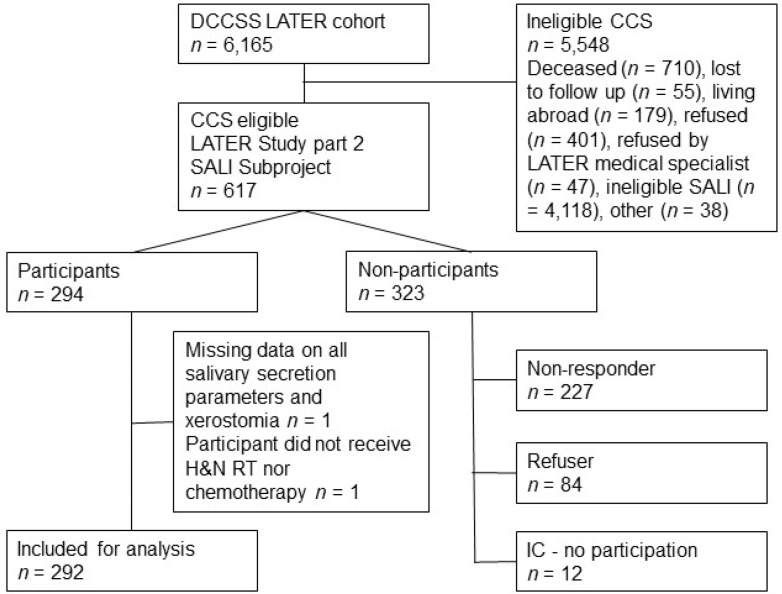
Flowchart of the DCCSS LATER 2 SALI study inclusion process. DCCSS LATER; Dutch Childhood Cancer Survivor Study Late Effects Study, IC—no participation; survivors signed informed consent for participation, however, data collection was hampered for different reasons. CCS who were ineligible for the SALI subproject were treated in outpatient clinics of DCCSS LATER 2, which were not participating in the SALI subproject.

**Table 1 cancers-14-03379-t001:** Patient and treatment-related characteristics.

Variable	Total *n* = 292 (100.0%)	Group 1: No H&N RT *n* = 198 (67.8%)	Group 2: H&N RT *n* = 94 (32.2%)	*p*
Gender				
Male	155 (53.1)	96 (48.5)	59 (62.8)	0.024 *
Female	137 (46.9)	102 (51.5)	35 (37.2)	
Diagnosis				
Hematological malignancy	216 (74.0)	147 (74.2)	69 (73.4)	<0.001 *
Brain tumor	19 (6.5)	1 (0.5)	18 (19.1)	
Solid tumor	57 (19.5)	50 (25.3)	7 (7.4)	
Age at enrollment (y)	32.15 (16.77–59.47)	29.93 (16.77–59.47)	39.34 (18.44–57.61)	<0.001 **
Age at cancer diagnosis (y)	5.20 (0.01–17.00)	3.97 (0.01–17.00)	7.47 (0.38–16.87)	<0.001 **
0 < 5	140 (47.9)	114 (57.6)	26 (27.7)	
5 < 10	89 (30.5)	47 (23.7)	42 (44.7)	
10 < 15	50 (17.1)	27 (13.6)	23 (24.5)	
>15	13 (4.5)	10 (5.1)	3 (3.2)	
Time since diagnosis (y)	25.26 (15.94–49.04)	23.96 (15.94–49.04)	32.61 (16.50–45.57)	<0.001 **
0 < 10	0 (0.0)	0 (0.0)	0 (0.0)	
10 < 20	64 (21.9)	51 (25.8)	13 (13.8)	
20 < 30	132 (45.2)	106 (53.5)	26 (27.7)	
>30	96 (32.9)	41 (20.7)	55 (58.5)	
Chemotherapy	281 (96.2)	198 (100.0)	83 (88.3)	<0.001 *
Alkylating agents	200 (68.5)	135 (68.2)	65 (69.1)	0.894 *
Vinca alkaloids	241 (82.5)	170 (85.9)	71 (75.5)	0.033 *
Epipodophyllotoxins	84 (28.8)	50 (25.3)	34 (36.2)	0.072 *
Anthracyclines	185 (63.4)	134 (67.7)	51 (54.3)	0.028 *
Platinum compounds	31 (10.6)	21 (10.6)	10 (10.6)	1.000 *
Antimetabolites	202 (69.2)	134 (67.7)	68 (72.3)	0.498 *
Radiotherapy	114 (39.0)	20 (10.1) ^a^	94 (100.0)	
H&N RT, to salivary glands			81 (86.2)	
H&N RT, not to salivary glands ^b^			6 (6.4)	
H&N RT to salivary glands unclear ^c^			7 (7.4)	
RT yes/no unclear ^d^		1 (0.5)		
Stem Cell Transplantation				
Autologous	11 (3.8)	4 (2.0)	7 (7.4)	<0.001 *
Allogeneic	38 (13.0)	16 (8.1)	22 (23.4)	
SCT unclear	1 (0.3)	1 (0.5)	0 (0.0)	
cGVHD	3 (1.0)	1 (0.5)	2 (2.1)	<0.001 *
Use of medication				
Unknown	7 (2.4)	6 (3.0)	1 (1.1)	0.004 *
≤3 medicines	249 (85.3)	176 (88.9)	73 (77.7)	
≥4 medicines	36 (12.3)	16 (8.1)	20 (21.3)	

H&N RT; head and neck radiotherapy. * Fisher’s exact test; ** Mann–Whitney U test. ^a^ No H&N RT. ^b^ Field of irradiation was the neck. ^c^ Field of irradiation was part of the brain. ^d^ It was unclear whether the participant received radiotherapy. Values are presented as *n* (column %) or median (range). Numbers do not always add up to 100% because of rounding.

**Table 2 cancers-14-03379-t002:** Applied radiation fields and dose of radiation (in Gy) in the group of CCS treated with H&N RT.

Radiation Field	Total *n* (%)	Radiation Dose	
		Mean (sd)	Median (Range)
Group 2: H&N RT	94 (100.0)	27.80 (17.91)	25.00 (5.00–100.80)
RT to salivary glands ^a^	81 (86.2)	25.17 (17.27)	24.00 (5.00–100.80)
Full brain	45	31.26 (11.99)	25.00 (18.00–55.70)
Face	8	46.69 (26.46)	37.80 (25.00–100.80)
TBI	29	9.33 (2.34)	8.00 (5.00–12.00)
RT not to salivary glands ^b^	6 (6.4)	33.20 (10.30)	39.70 (19.80–40.00)
Unclear RT to salivary glands ^c^	7 (7.4)	53.57 (3.58)	54.00 (50.40–60.00)

H&N RT; head/neck radiotherapy, TBI; total body irradiation. ^a^ One CCS received both TBI and RT to the brain. These dose values have been summed. ^b^ Field of irradiation was the neck. ^c^ Field of irradiation was part of the brain.

**Table 3 cancers-14-03379-t003:** Prevalence of xerostomia, unstimulated and stimulated salivary flow rate, stratified according to the type of treatment and gender in childhood cancer survivors.

Variable	Total	Group 1: No H&N RT	Group 2: H&N RT	Subgroup 2A: RT to Salivary Glands ^e^	*p* ^¥^	Male	Female	*p* ^µ^
Xerostomia inventory (XI) (*n*)	*n* = 233	*n* = 156	*n* = 77	*n* = 65		*n* = 125	*n* = 108	
Median XI-score (range) ^a^	17.00 (11.00–41.00)	17.00 (11.00–39.00)	16.00 (11.00–41.00)	17.00 (11.00–36.00)	0.980 **	16.00 (11.00–39.00)	17.00 (11.00–41.00)	0.174 **
Mean XI-score (sd) ^a^	17.77 (6.01)	17.80 (5.79)	17.71 (6.48)	17.75 (5.71)		17.20 (5.56)	18.44 (6.46)	
Number with xerostomia (%) ^b^	22 (9.4)	13 (8.3)	9 (11.7)	7 (10.8)	0.444 *	10 (8.0)	12 (11.1)	0.502 *
UWS (*n*)	*n* = 269	*n* = 184	*n* = 85	*n* = 73		*n* = 144	*n* = 125	
Median (range) ^c^	0.29 (0.00–1.42)	0.31 (0.03–1.36)	0.21 (0.00–1.42)	0.21 (0.00–1.42)	0.003 **	0.31 (0.00–1.42)	0.25 (0.01–1.36)	0.007 **
Mean (sd) ^c^	0.35 (0.25)	0.37 (0.25)	0.30 (0.27)	0.31 (0.28)		0.39 (0.27)	0.30 (0.22)	
Hyposalivation < 0.2 mL/min ^d^	86 (32.0)	47 (25.5)	39 (45.9)	34 (46.6)	0.002 *	37 (25.7)	49 (39.2)	0.019 *
Severe hyposalivation < 0.1 mL/min ^d^	24 (8.9)	11 (6.0)	13 (15.3)	10 (13.7)	0.074 *	8 (5.6)	16 (12.8)	0.052 *
SWS (*n*)	*n* = 271	*n* = 186	*n* = 85	*n* = 73		*n* = 146	*n* = 125	
Median (range) ^c^	0.98 (0.01–4.03)	1.11 (0.01–3.35)	0.78 (0.01–4.03)	0.78 (0.01–4.03)	0.003 **	1.14 (0.01–4.03)	0.83 (0.01–3.35)	0.001 **
Mean (sd) ^c^	1.14 (0.71)	1.20 (0.69)	0.99 (0.73)	0.98 (0.73)		1.27 (0.76)	0.98 (0.61)	
Hyposalivation < 0.7 mL/min ^d^	86 (31.7)	48 (25.8)	38 (44.7)	33 (45.2)	0.004 *	37 (25.3)	49 (39.2)	0.018 *
Severe hyposalivation < 0.5 mL/min ^d^	38 (14.0)	21 (11.3)	17 (20.0)	13 (17.8)	0.219 *	13 (8.9)	25 (20.0)	0.013 *

UWS; unstimulated whole salivary flow rate, SWS; stimulated whole salivary flow rate. Numbers of XI-scores, UWS values, and SWS values were different as not all parameters were measured in each individual. ^¥^ *p* value of the analysis between Group 1 (participant with unknown information about radiotherapy was excluded from analysis) and the subgroup: RT to salivary glands. ^µ^ *p* value of the analysis between males and females. ^a^ XI scores are presented [23]. ^b^ Values are presented as *n* (%) of participants that had an XI score above 25. ^c^ Values are presented as mL/min ^d^ Values are presented as *n* (%) of participants who had hyposalivation. ^e^ This subgroup is part of Group 2. Survivors were excluded from this subgroup when information on radiotherapy to salivary glands was unclear (*n* = 6) or salivary glands did not receive irradiation (*n* = 6). * Fisher’s exact test. ** Mann–Whitney U test.

**Table 4 cancers-14-03379-t004:** Association between objective stimulated and unstimulated salivary flow measurements and xerostomia (data are presented as *n* (total %)).

**Variable**	**UWS, >0.2 mL/min**	**UWS, ≤0.2 mL/min**	**Total**	** *p ** **
SWS, >0.7 mL/min	150 (55.8)	34 (12.6)	184 (68.4)	<0.001
≤0.7 mL/min	33 (12.3)	52 (19.3)	85 (31.6)	
Total	183 (68.0)	86 (32.0)	269 (100.0)	
	**No xerostomia**	**Xerostomia**	**Total**	** *p ** **
UWS, >0.2 mL/min	129 (61.4)	12 (5.7)	141 (67.1)	0.332
≤0.2 mL/min	60 (28.6)	9 (4.3)	69 (32.9)	
Total	189 (90.0)	21 (10.0)	210 (100.0)	
	**No xerostomia**	**Xerostomia ^a^**	**Total**	** *p ** **
SWS, >0.7 mL/min	133 (62.7)	15 (7.1)	148 (69.8)	1.000
≤0.7 mL/min	57 (26.9)	7 (3.3)	64 (30.2)	
Total	190 (89.6)	22 (10.4)	212 (100.0)	

UWS; unstimulated whole salivary flow rate, SWS; stimulated whole salivary flow rate. ^a^ Values are presented as *n* (%) of participants that had an XI score above 25. * Chi-square test.

**Table 5 cancers-14-03379-t005:** Poisson regression analysis for hyposalivation, measured by unstimulated whole salivary flow rates in childhood cancer survivors (*n* = 257).

Variable	Number of Survivors ^a^	Hyposalivation < 0.2 mL/min (*n*)	Risk Ratio	95% CI	*p*
Gender					
Male	141	37	1.0 (ref)		
Female	116	46	1.52	1.06 to 2.19	0.023
Age at diagnosis (per 1 year increase)	257	83	1.01	0.97 to 1.04	0.805
Time since diagnosis (per 10 year increase)	257	83	1.42	1.15 to 1.75	0.001
Radiotherapy dose (Gy) to salivary glands					
0 Gy ^b^	184	49	1.0 (ref)		
>0 and ≤12 Gy	26	10	1.48	0.89 to 2.47	0.128
>12 and ≤34 Gy	32	16	1.31	0.83 to 2.07	0.240
>34 Gy	15	8	2.10	1.21 to 3.63	0.008
Number of medications (per 1 number increase)	257	83	1.02	0.94 to 1.11	0.621

Gy: gray, (ref): reference category. ^a^ 12 participants were excluded from this analysis due to unclear information about: radiotherapy (*n* = 1), radiotherapy to salivary glands (*n* = 6) and the number of medication (*n* = 5). ^b^ Group of RT dose of 0 Gray consists of Group 1: chemotherapy (*n* = 178) and survivors who did receive H&N RT but not to salivary glands (*n* = 6).

**Table 6 cancers-14-03379-t006:** Poisson regression analysis for hyposalivation, measured by stimulated whole salivary flow rates in childhood cancer survivors (*n* = 264).

Variable	Number of Survivors ^a^	Hyposalivation < 0.7 mL/min (*n*)	Risk Ratio	95% CI	*p*
Gender					
Male	144	37	1.0 (ref)		
Female	120	45	1.57	1.10 to 2.23	0.013
Age at diagnosis (per 1 year increase)	264	82	0.98	0.94 to 1.03	0.442
Time since diagnosis (per 10 year increase)	264	82	0.91	0.71 to 1.16	0.433
Radiotherapy dose to salivary glands					
0 Gy ^b^	191	49	1.0 (ref)		
>0 and ≤12 Gy	26	10	1.74	0.95 to 3.20	0.073
>12 and ≤34 Gy	32	14	2.15	1.26 to 3.67	0.005
>34 Gy	15	9	2.25	1.35 to 3.76	0.002
Chemotherapy					
No vinca alkaloids	39	19	1.0 (ref)		
Vinca alkaloids	225	63	0.67	0.44 to 1.02	0.059
No anthracyclines	91	36	1.0 (ref)		
Anthracyclines	173	46	0.81	0.56 to 1.18	0.281
No alkylating agents	80	32	1.0 (ref)		
Alkylating agents	184	50	0.81	0.55 to 1.19	0.282

Gy: gray, (ref): reference category; ^a^ 7 participants were excluded from this analysis due to unclear information about radiotherapy (*n* = 1) and radiotherapy to salivary glands (*n* = 6). ^b^ Group of RT dose of 0 Gray consists of Group 1: chemotherapy (*n* = 185) and survivors who did receive H&N RT but not to salivary glands (*n* = 6).

## Data Availability

The data that support the findings of this study will be stored for at least 10 years. Request for data can be made via TDC LATER, with an application of intent (M.vanderHeiden@prinsesmaximacentrum.nl).

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
