# Peer review of "Prevalence and Risk Factors for Hyposalivation and Xerostomia in Childhood Cancer Survivors Following Different Treatment Modalities—A Dutch Childhood Cancer Survivor Study Late Effects 2 Clinical Study (DCCSS LATER 2)"

_cancers, 2022, doi:10.3390/cancers14143379_

Round 1

Reviewer 1 Report

This study is important as it addresses the risk factors for xerostomia in pediatrics. However, I do not think that dividing the cohort into 2 groups was a good approach as each group is significantly different in terms of patient and disease characteristics. Also, more details about the treatment modalities will be useful. I would also recommend mentioning the RT dose to each salivary gland separately, especially parotid and submandibular glands. 

Author Response

Dear reviewer,

We thank you for thoroughly reading our manuscript and the valuable comments that helped us to improve our paper. We have looked carefully at each of the topics raised and we have modified our manuscript accordingly. Please find below a point-by-point reaction. 

We feel that these changes have improved the manuscript considerably. We hope that this revised version is now acceptable for publication in Cancers. For your convenience, we have also added an additional version of the manuscript with track-changes.

We look forward to your response,

With kind regards on behalf of all authors,

Juliette Stolze

Review report form 1

Open Review

English language and style

( ) Extensive editing of English language and style required
( ) Moderate English changes required
( ) English language and style are fine/minor spell check required
(x) I don't feel qualified to judge about the English language and style

Yes

Can be improved

Must be improved

Not applicable

Does the introduction provide sufficient background and include all relevant references?

( )

( )

(x)

( )

Are all the cited references relevant to the research?

( )

(x)

( )

( )

Is the research design appropriate?

( )

( )

(x)

( )

Are the methods adequately described?

( )

(x)

( )

( )

Are the results clearly presented?

( )

(x)

( )

( )

Are the conclusions supported by the results?

( )

(x)

( )

( )

Comments and Suggestions for Authors

  • This study is important as it addresses the risk factors for xerostomia in pediatrics. However, I do not think that dividing the cohort into 2 groups was a good approach as each group is significantly different in terms of patient and disease characteristics. Also, more details about the treatment modalities will be useful.

Thank you for your comment.

Since our hypothesis was that radiotherapy and chemotherapy would affect long term salivary gland function to a different extent, we decided to focus on these treatments in this manuscript due to the lack of research in this field.

With Table 1, we want to be transparent on the differences in patient and treatment characteristics between survivors who received H&N RT and those who did not receive H&N RT. Though we divided the cohort into 2 groups (H&N RT and no H&N RT) and a subgroup (RT to salivary glands) for description of the results, we did not use these groups in multivariable analysis or risk analysis. Multivariable analysis was performed with the total group for hyposalivation based on unstimulated and stimulated salivary flow rate. In univariable analysis, hyposalivation and xerostomia were compared between the no-H&N group and the subgroup 2A of the H&N RT group.

We added an explanation in the methods section and tried to clarify why we divided the study sample to different groups. 

Since our hypothesis was that radiotherapy and chemotherapy would affect long-term salivary gland function to a different extent, during inclusion for the SALI subproject, we aimed to create three study groups: a group of CCS who received head and neck radiotherapy (H&N RT) or total body irradiation (TBI) without chronic Graft versus Host Disease (cGVHD), a group of CCS with (a history of) cGVHD after HSCT, and a group of CCS treated with chemotherapy and no H&N RT or TBI.

Thereby, we added a few words to emphasize that all CCS of the No H&N RT group received chemotherapy.

Therefore, this group was omitted from the study, resulting in two groups for analysis: Group 1: CCS who did not receive H&N RT (but all received chemotherapy), Group 2: CCS who received H&N RT and/or TBI.

For information about the treatment modalities, we assume that we have mentioned the most important details in the methods section.

  • I would also recommend mentioning the RT dose to each salivary gland separately, especially parotid and submandibular glands. 

We understand that you prefer more clarity on the dosage to each of the salivary glands. However, the available data do not allow such specified information.

Our data concerned different radiation fields of the head, what has been compiled by our radiology specialist. Radiotherapy was administered to different regions: a) field 1, 2, 3 and 4 (full brain irradiation); b) field 1 and 2 OR field 3 and 4 (partial brain irradiation); c) field 5, 6 and 7 (face).

We determined whether salivary glands were within these fields of radiation (a, b or c) by use of anatomy and literature of location of the salivary glands. The field ‘full brain irradiation’ includes the parotid glands and the field ‘face’ includes submandibular and sublingual glands and possibly the parotid glands. If the patient received full brain RT or RT to the face; this was assessed as “H&N RT to salivary glands”. Of the field ‘partial brain irradiation’, it is unclear whether parotid glands were in the field of irradiation (in case fields 3 and 4 were irradiated). This was assessed as “H&N RT to salivary glands unclear”. If RT was administered to the neck but not to one of the fields 1 to 7 of the head; this was assessed as “H&N RT not to salivary glands”.

In table 2, applied radiation fields and dose of radiation are shown. With this information, it is possible to deduct the dose to each gland in table 2. Full brain RT include parotid glands. Face RT include sublingual and submandibular glands and possibly parotid glands. However, we think it is not wise to rename the sections into specific glands, as our data is not specified enough.

We tried to clarify our methodology by adjusting some areas in the manuscript, see below.

Patients who received H&N RT (Group 2) were further subdivided into whether they did (subgroup 2A) or did not (subgroup 2B) receive radiotherapy to the salivary glands based on coded radiation fields in the LATER data registry. The radiation fields concerned full brain RT, partial brain RT, the face and the neck. Whether salivary glands were within these fields of radiation, was determined by use of anatomy and literature of the location of the salivary glands. In CCS who received RT to the face (including the sublingual and submandibular glands and possibly the parotid glands) and/or the central part of the head (including the parotid gland), radiotherapy was classified as “radiotherapy to salivary gland”. In CCS who received partial brain RT, it was unclear whether salivary glands were in the field of irradiation, thus radiotherapy was classified as “radiotherapy to salivary glands unclear”. Radiotherapy to the neck was classified as “radiotherapy not to salivary glands”. If information about the radiation fields was unclear, these CCS were not included in one of the two subgroups (2A or 2B) and thus excluded from analysis.

Thereby, we replaced the location of the superscript b, c, d to the left column of Table 1. We also added a superscript c to table 2. For these changes, we would like to refer to the manuscript.

Submission Date

31 May 2022

Date of this review

13 Jun 2022 17:32:59

Reviewer 2 Report

The study presents valuable data. Few points needs to be addressed before considering to publication

Abstract

1.The sentence  Participants filled out the Xerostomia Inventory (XI).” Is not required. OR values should be provided when mentioning risk factors.

2.The sentence “Hyposalivation was present in 32.0% (UWS) and 31.7% (SWS) of CCS” doesn’t make sense.

3.The prevalence of hyposalivation (whats the prevalence?) was “much higher”?? How much??? than the prevalence of xerostomia (9.4%). Is it significant?

4. Abstract does not highlight the main results.

5. Abstract is not properly structured. Please re write the abstract.

Discussion

1. The implication of this results need to be explored and critically reviewed.

3. Formatting issues

1.      The headings need to be formatted eg discussion is starting with small letter d

2.      Supplementary material is placed under conclusion part

Author Response

Dear reviewer,

We thank you for thoroughly reading our manuscript and the valuable comments that helped us to improve our paper. We have looked carefully at each of the topics raised and we have modified our manuscript accordingly. Please find below a point-by-point reaction. 

We feel that these changes have improved the manuscript considerably. We hope that this revised version is now acceptable for publication in Cancers. For your convenience, we have also added an additional version of the manuscript with track-changes.

We look forward to your response,

With kind regards on behalf of all authors,

Juliette Stolze

Review report 2

Open Review

English language and style

( ) Extensive editing of English language and style required
(x) Moderate English changes required
( ) English language and style are fine/minor spell check required
( ) I don't feel qualified to judge about the English language and style

Yes

Can be improved

Must be improved

Not applicable

Does the introduction provide sufficient background and include all relevant references?

(x)

( )

( )

( )

Are all the cited references relevant to the research?

(x)

( )

( )

( )

Is the research design appropriate?

(x)

( )

( )

( )

Are the methods adequately described?

(x)

( )

( )

( )

Are the results clearly presented?

( )

(x)

( )

( )

Are the conclusions supported by the results?

( )

(x)

( )

( )

Comments and Suggestions for Authors

The study presents valuable data. Few points needs to be addressed before considering to publication

Abstract

1.The sentence  “Participants filled out the Xerostomia Inventory (XI).” Is not required. OR values should be provided when mentioning risk factors.

2.The sentence “Hyposalivation was present in 32.0% (UWS) and 31.7% (SWS) of CCS” doesn’t make sense.

3.The prevalence of hyposalivation (whats the prevalence?) was “much higher”?? How much??? than the prevalence of xerostomia (9.4%). Is it significant?

  1. Abstract does not highlight the main results.
  2. Abstract is not properly structured. Please re write the abstract.

We thank the reviewer for these suggestions. We adjusted the abstract according to the suggestions. The abstract now consists a maximum of 200 words.

Background: Limited data are available on risk factors of salivary gland dysfunction in long-term childhood cancer survivors (CCS). The objective of this cross-sectional study, part of the multidisciplinary multicenter Dutch CCS Study Late Effects 2 (DCCSS LATER-2), is to assess the prevalence of and risk factors for hyposalivation and xerostomia in CCS. Methods: From February 2016 until March 2020, 292 CCS were included. Data with regard to gender, age at study, diagnosis, age at diagnosis and treatment characteristics were collected, as well as unstimulated (UWS) and stimulated whole salivary flow rate (SWS). Xerostomia was assessed with the Xerostomia Inventory (XI) questionnaire. Multivariable Poisson regression analyses were used to evaluate the association between potential risk factors and occurrence of hyposalivation. Results: The minimum time between diagnosis and study enrollment was 15 years. The prevalence of hyposalivation was 32% and the prevalence of xerostomia was 9.4%. Hyposalivation and xerostomia were not significantly correlated. Risk factors for hyposalivation were female gender and a higher dose of radiotherapy (>12 Gy) to the salivary gland region. Conclusion: Considering the importance of saliva for oral health, screening for hyposalivation in CCS is suggested in order to provide optimal oral supportive care aimed to improve oral health.

Discussion

  1. The implication of this results need to be explored and critically reviewed.

We thank the reviewer for this comment. We expanded the discussion to include the implications and consequences of the results.

In this study (3), xerostomia was assessed as positive answers to nine questions indicating dry mouth, summed to determine the xerostomia score. The study found that the xerostomia score was inversely correlated to UWS and SWS (p<0.01) (3).

In the present study, however, symptoms of xerostomia did not correlate with the occurrence of, objectively measured, hyposalivation. Only a small group of the participants who had objectively decreased salivary secretion rate experienced xerostomia (10% to 20%). This is an important finding, indicating that not all CCS recognize their actual decreased salivary secretion rate. This may be related to the composition of saliva. Xerostomia merely affects the oral health related quality of life, whereas hyposalivation directly affects the oral health by increasing the risk on caries, gingival inflammation, oral discomfort and pain, dental erosion and oral mucosal infections. According to our results, it appears that CCS are unaware of the presence of hyposalivation. As hyposalivation is prevalent among CCS and negatively affects the oral health, it deserves more attention. This is especially important in CCS, because hyposalivation and xerostomia may develop already at a young age and therefore they may not appreciate their oral mouth as abnormal and may not complain of xerostomia. Thus, we recommend that any evaluation of dry mouth should not be limited to assessment of subjective complaints, but should also include measurement of salivary secretion rates.

We have tried to make the discussion more concise in a few paragraphs and therefore adjusted or added a few sentences. You can see these changes in the word document with ‘track changes’ function.

Thereby, we slightly adjusted the conclusion to remain critical.

Therefore, to reveal this late effect, screening of salivary gland dysfunction during long-term follow up in CCS is recommended in order to provide optimal oral supportive care aimed to improve oral health. We would like to make the suggestion that late effects screening guidelines could be adjusted accordingly.

  1. Formatting issues
  2. The headings need to be formatted eg discussion is starting with small letter d
  3. Supplementary material is placed under conclusion part

We adjusted the issues as mentioned by the reviewer.

Submission Date

31 May 2022

Date of this review

08 Jun 2022 04:55:50

Reviewer 3 Report

The authors conducting a Cross-sectional study aimed to assess the prevalence and risk factors of hyposalivation and xerostomia among Dutch childhood Cancer Survivors. The Study is well conducting and written.

I have some minor comments:

Title: according to the journal instruction (https://www.mdpi.com/journal/cancers/instructions), the title should be free of abbreviation. The abbreviation DCCSS should be removed or replaced.

Abstract: please add an introductionary sentence before the objective 

Introduction: The impact of hyposalivation and/or xerostomia on Oral Health-related Quality of Life is very essential and should be mentioned in the introduction in details

Methods: you mentioned  "Informed consent was obtained from all subjects."  

as your study subjects may includes childs, do you collect the consent from in this case from the parents ?

Study groups: please add explanation why you divided the study sample to the mentioned groups.

General comment: do you assess also the teeth development and the caries prevalence among the study sample? it would be great to publish such data in your collective .

Best regards

Author Response

Dear reviewer,

We thank you for thoroughly reading our manuscript and the valuable comments that helped us to improve our paper. We have looked carefully at each of the topics raised and we have modified our manuscript accordingly. Please find below a point-by-point reaction. 

We feel that these changes have improved the manuscript considerably. We hope that this revised version is now acceptable for publication in Cancers. For your convenience, we have also added an additional version of the manuscript with track-changes.

We look forward to your response,

With kind regards on behalf of all authors,

Juliette Stolze

Reviewer report 3

Open Review

English language and style

( ) Extensive editing of English language and style required
(x) Moderate English changes required
( ) English language and style are fine/minor spell check required
( ) I don't feel qualified to judge about the English language and style

Yes

Can be improved

Must be improved

Not applicable

Does the introduction provide sufficient background and include all relevant references?

( )

(x)

( )

( )

Are all the cited references relevant to the research?

( )

(x)

( )

( )

Is the research design appropriate?

(x)

( )

( )

( )

Are the methods adequately described?

( )

(x)

( )

( )

Are the results clearly presented?

(x)

( )

( )

( )

Are the conclusions supported by the results?

(x)

( )

( )

( )

Comments and Suggestions for Authors

The authors conducting a Cross-sectional study aimed to assess the prevalence and risk factors of hyposalivation and xerostomia among Dutch childhood Cancer Survivors. The Study is well conducting and written.

I have some minor comments:

  • Title: according to the journal instruction (https://www.mdpi.com/journal/cancers/instructions), the title should be free of abbreviation. The abbreviation DCCSS should be removed or replaced.

Thank you for the comment. We adjusted our title to Prevalence and risk factors for hyposalivation and xerostomia in childhood cancer survivors following different treatment modalities - a Dutch Childhood Cancer Survivor Study Late Effects 2 Clinical Study (DCCSS LATER-2).

  • Abstract: please add an introductionary sentence before the objective 

We agree with the reviewer and added an introductionary sentence before the objective. Thereby, we adjusted the abstract slightly.

Background. Limited data are available on risk factors of salivary gland dysfunction in long-term childhood cancer survivors (CCS). The objective of this cross-sectional study, part of the multidisciplinary multicenter Dutch CCS Study Late Effects 2 (DCCSS LATER-2), is to assess the prevalence of and risk factors for hyposalivation and xerostomia in CCS. Methods. From February 2016 until March 2020, 292 CCS were included. Data with regard to gender, age at study, diagnosis, age at diagnosis and treatment characteristics were collected, as well as unstimulated (UWS) and stimulated whole salivary flow rate (SWS). Xerostomia was assessed with the questionnaire Xerostomia Inventory (XI). Multivariable Poisson regression analyses were used to evaluate the association between potential risk factors and occurrence of hyposalivation. Results. The minimum time between diagnosis and study enrollment was 15 years. The prevalence of hyposalivation was 32% and the prevalence of xerostomia was 9.4%. Hyposalivation and xerostomia were not significantly correlated. Risk factors for hyposalivation were female gender and a higher dose of radiotherapy (>12 Gy) to the salivary gland region. Conclusion. Considering the importance of saliva for oral health, screening for hyposalivation in CCS is suggested in order to provide optimal oral supportive care aimed to improve oral health.

  • Introduction: The impact of hyposalivation and/or xerostomia on Oral Health-related Quality of Life is very essential and should be mentioned in the introduction in details

We agree with the reviewer on this point and added a sentence in the introduction in which we emphasize the impact of salivary gland dysfunction on OHRQoL.

Complications arising from hyposalivation include oral discomfort and pain, difficulty with speaking, mastication and swallowing, decreased taste perception, nutritional compromise, sleep disorders, dental erosion, xerostomia, increased prevalence of dental caries and susceptibility to oral mucosal infections (7,8). These complications can significantly affect oral health-related quality of life (9,10).

We plan to address the topic of hyposalivation and oral OHRQoL more extensively in a separate paper.

  • Methods: you mentioned  "Informed consent was obtained from all subjects." As your study subjects may includes childs, do you collect the consent from in this case from the parents ?

This study does not include children, it does include survivors from childhood cancer. The youngest participant is 16.77 years old. The age at enrollment of all parcipants is described in table 1. Our informed consent is thus collected by the participants themselves.

  • Study groups: please add explanation why you divided the study sample to the mentioned groups.

Thank you for your comment.

Since our hypothesis was that radiotherapy and chemotherapy would affect long term salivary gland function to a different extent, we decided to focus on these treatments in this manuscript due to the lack of research in this field.

With Table 1, we want to be transparent on the differences in patient and treatment characteristics between survivors who received H&N RT and those who did not receive H&N RT. Though we divided the cohort into 2 groups (H&N RT and no H&N RT) and a subgroup (RT to salivary glands) for description of the results, we did not use these groups in multivariable analysis or risk analysis. Multivariable analysis was performed with the total group for hyposalivation based on unstimulated and stimulated salivary flow rate. In univariable analysis, hyposalivation and xerostomia were compared between the no-H&N group and the subgroup 2A of the H&N RT group.

We added an explanation in the methods section and tried to clarify why we divided the study sample to different groups. 

Since our hypothesis was that radiotherapy and chemotherapy would affect long-term salivary gland function to a different extent, during inclusion for the SALI subproject, we aimed to create three study groups: a group of CCS who received head and neck radiotherapy (H&N RT) or total body irradiation (TBI) without chronic Graft versus Host Disease (cGVHD), a group of CCS with (a history of) cGVHD after HSCT, and a group of CCS treated with chemotherapy and no H&N RT or TBI.

Thereby, we added a few words to emphasize that all CCS of the No H&N RT group received chemotherapy.

Therefore, this group was omitted from the study, resulting in two groups for analysis: Group 1: CCS who did not receive H&N RT (but all received chemotherapy), Group 2: CCS who received H&N RT and/or TBI.

  • General comment: do you assess also the teeth development and the caries prevalence among the study sample? it would be great to publish such data in your collective .

Thank you for your recommendation. We already performed a study about dental developmental disorders and oral health (i.e. caries prevalence) assessed by surveys among dentists of the included CCS. The number of included CCS in that study concerns approximately 70% of the study inclusion in the present study. We kindly refer to our previously published paper in Cancers (https://doi.org/10.3390/cancers13215264

Submission Date

31 May 2022

Date of this review

16 Jun 2022 10:51:38
